# Trans-Cinnamaldehyde Exhibits Synergy with Conventional Antibiotic against Methicillin-Resistant *Staphylococcus aureus*

**DOI:** 10.3390/ijms22052752

**Published:** 2021-03-09

**Authors:** Shu Wang, Ok-Hwa Kang, Dong-Yeul Kwon

**Affiliations:** Department of Oriental Pharmacy, College of Pharmacy and Wonkwang Oriental Medicines Research Institute, Wonkwang University, Iksan 54538, Jeonbuk, Korea; wshu1996@gmail.com

**Keywords:** MRSA, trans-cinnamaldehyde, β-lactam antibiotics, synergy, *mecA*, PBP2a, biofilm

## Abstract

Methicillin-resistant *Staphylococcus aureus* (MRSA) is a major nosocomial pathogen worldwide and has acquired multiple resistance to a wide range of antibiotics. Hence, there is a pressing need to explore novel strategies to overcome the increase in antimicrobial resistance. The present study aims to investigate the efficacy and mechanism of plant-derived antimicrobials, trans-cinnamaldehyde (TCA) in decreasing MRSA’s resistance to eight conventional antibiotics. A checkerboard dilution test and time–kill curve assay are used to determine the synergistic effects of TCA combined with the antibiotics. The results indicated that TCA increased the antibacterial activity of the antibiotics by 2-16-fold. To study the mechanism of the synergism, we analyzed the *mecA* transcription gene and the penicillin-binding protein 2a level of MRSA treated with TCA by quantitative RT-PCR or Western blot assay. The gene transcription and the protein level were significantly inhibited. Additionally, it was verified that TCA can significantly inhibit the biofilm, which is highly resistant to antibiotics. The expression of the biofilm regulatory gene *hld* of MRSA after TCA treatment was also significantly downregulated. These findings suggest that TCA maybe is an exceptionally potent modulator of antibiotics.

## 1. Introduction

Gram-positive bacterium methicillin-resistant *Staphylococcus aureus* (MRSA) is a prominent human pathogen worldwide [1], which is a major cause of antibiotic-resistant healthcare-associated infections. In recent years, community-acquired strains of infection have appeared and have begun to spread among healthy persons [2]. MRSA resists most antibiotics, owing to the strain that produces the penicillin-binding protein 2a (PBP2a). The PBP2a is a modified penicillin-binding protein encoded by the *mecA* gene, demonstrating reduced affinity to β-lactam antibiotics. *MecA* is regulated by genes *mecI* and *mecR1* located on the bacterial chromosome [3]. In the past, the antibiotics commonly used to treat MRSA were vancomycin and oral linezolid. However, moderate resistance to vancomycin or linezolid has emerged [4], posing an urgent demand for novel antibacterial substances or therapeutic options.

Natural products have been a rich source of compounds with structural and chemical diversity for drug discovery [5]. Since plant antibacterial agents contain different functional groups, their antibacterial activity can be attributed to multiple mechanisms. Therefore, compared with antibiotics, bacteria are relatively less likely to develop resistance to plant-derived antibacterial substances [6]. Cinnamon is a traditional spice, which is widely used around the world [7]. It has also been used since ancient times as a food preservative by acting against foodborne pathogens and spoilage bacteria [8]. Cinnamon extract components have demonstrated a range of medical properties, cinnamic acid possesses antibacterial activity based on directed self-assembly [9]; procyanidin C1 acts as a potential insulin sensitizer that targets adipocytes [10]; trans-cinnamaldehyde (TCA) mitigates intestinal inflammation [11]. TCA, the main component of cinnamon, is an aromatic aldehyde existing in cinnamon bark extract. It is a food-grade molecule and is listed as a recognized safe agent by the Food and Drug Administration [12]. In recent years, there are various studies on TCA as a natural antibacterial substance, with antibacterial activity against *Listeria monocytogenes*, *Cronobacter* spp., and MRSA [13].

In this study, we evaluated the synergistic antibacterial activity of TCA combined with eight conventional antibiotics and investigated the mechanism at the molecular level.

## 2. Results

### 2.1. Synergistic Antibacterial Activity

A broth microdilution assay was used to determine the minimal inhibitory concentration (MIC) of TCA, and the value against all strains (reference strain ATCC 33591 and two clinical isolates) was 250 μg/mL. The results of checkerboard dilution are indicated in Table 1. For all studied strains, the fractional inhibitory concentration index (FICI) of TCA combinations with conventional antibiotics (including ampicillin, oxacillin, gentamicin, vancomycin, amoxicillin, ceftazidine, amikacin, and cefoxitin) were 0.19–0.1, showing synergy, partial synergy, and additive effect. TCA combined with amikacin had the best synergism, with a mean FICI of 0.27. TCA combined with gentamicin, vancomycin, and amoxicillin had a significant synergistic effect. The remaining combinations, including those with the antibiotics, ampicillin, oxacillin, ceftazidine, and cefoxitin showed relatively less significant interactions with TCA against MRSA, although the MIC value of antibiotics also reduced by at least 2-fold against all three strains.

### 2.2. Time–Kill Assay

According to the results of the checkerboard assays, three antibiotics with excellent synergistic effect with TCA were selected to further explore the synergistic effect by a time–kill assay. The use of antibiotics with 1/2 MIC alone had no effect on the growth of bacteria after 24 h. In all combination groups, bacterial growth was detected to be inhibited after 4 h (Figure 1a–c). TCA (1/2 MIC) combined with three antibiotics (1/2 MIC), amikacin, gentamicin, and oxacillin in the treatment of *S. aureus*, ATCC 33591, DPS-1, and DPS-3, displayed significantly synergistic interactions, respectively, and more than 3 log10 reductions in colony count after 24 h. Besides, a combination of TCA (1/2 MIC) and gentamicin (1/2 MIC) exhibited bactericidal activity against MRSA (DPS-1) after overnight incubation.

### 2.3. TCA Represses the Transcription of MecA, MecR1, and MecI in S. aureus

The transcriptional levels of *mecA*, *mecR1,* and *mecI* were inhibited in *S. aureus* upon the treatment with 1/8 MIC concentrations of TCA, and the transcription levels of three genes were affected by the treatment with graded subinhibitory concentrations (Figure 2). In the presence of 1/2 MIC of TCA, the transcriptional levels of *mecA*, *mecR1,* and *mecI* were decreased by 1.8-fold, 2.3-fold, and 2.5-fold, respectively.

### 2.4. Expression of PBP2a in MRSA Treated with TCA

Western blot was used to evaluate the expression level of PBP2A in TCA treated or untreated MRSA strains (Figure 3). The PBP2a levels were significantly reduced after treatment with TCA of sub-concentrations (1/8 MIC, 1/4 MIC, and 1/2 MIC), and TCA caused a concentration-dependent decrease in the expression of PBP2a. Moreover, the PBP2a production of *S. aureus* strains was almost undetectable after exposure to 1/2 MIC of TCA. The expression levels of PBP2A in MRSA decreased after TCA addition, corresponding to the low expression of *mecA* transcription after primer extension.

### 2.5. Biofilm Inhibitory Assay and the Transcription of hld in S. aureus

The biofilm formation of the two strains (mean of 1 isolates ATCC 33591 and 1 standard isolate DPS-1) in the presence of sub-concentration of TCA was inhibitory significantly, and TCA caused a concentration-dependent inhibitory effect on the biofilm formation (Figure 4a). TCA also significantly down-regulated the expression of the biofilm regulatory gene *hld* of MRSA. In the presence of 1/2 MIC of TCA, the transcriptional levels of *hld* were reduced by 5.5-fold or 5-fold on ATCC 33591 or DPS-1, respectively (Figure 4b). 

## 3. Discussion

With the slow development of novel drugs, a combination of established drugs that restore therapeutic effects through synergistic interactions is a promising method to overcome the increasing antibiotic resistance [3]. In the present study, the synergistic anti-MRSA effects and mechanism of the natural antibacterial agent TCA and eight conventional antibiotics including β-lactam antibiotics were evaluated for the first time. The study pointed out that TCA had a potent potential to reverse the sensitivity of conventional antibiotics to MRSA. The results of checkerboard dilution indicated that TCA had synergistic or partial synergistic effects with all the eight conventional antibiotics tested, especially in combination with amikacin, which reduced the MIC of amikacin against MRSA by 16-fold and the FICI value was as low as 0.19. According to the standard of the Clinical and Laboratory Standards Institute (CLSI), the clinical MRSA strain DPS-3 is an oxacillin-resistant strain. The MIC of oxacillin is reduced by 8-fold when combined with TCA against MRSA. TCA significantly improved the antibacterial sensitivity of oxacillin. However, the MIC of oxacillin is only reduced by 2-fold when the TCA combined with oxacillin against clinical DPS-1. This may be due to the differential expression of the *mecA* gene in different clinical strains.

Time–kill assay was employed to further evaluate the synergy effect of TCA combined with conventional antibiotics, and three results with excellent synergy effect were listed in the present study. The results demonstrated that compared with treatment with antibiotics or TCA alone at sub-MIC concentration, the two-drug combination groups exhibited a significant synergy effect and more than 3 log10 reduction in colony count after 24 h, respectively. TCA combined with gentamicin in sub-MIC can completely inhibit the growth of MRSA even after 24 h. In addition, TCA significantly inhibited MRSA growth after 4 h, which may mean that combination therapy inhibits the bacteria faster than the traditional single antibiotic therapy.

We supported that the synergy mechanism between TCA and antibiotics was multifaceted. First, the results of qRT-PCR and Western blot revealed that TCA inhibited *mecA* and PBP2a in a dose-dependent manner at sub-concentrations, suggesting that the synergistic effect of TCA and β-lactam antibiotics was caused by the fact that TCA downregulated the transcription and translation of antibiotic resistance gene *mecA*. Significantly, synergy was also presented when TCA was combined with non-beta lactam antibiotics. We speculated that was because TCA can destroy the biofilm and cell membrane of MRSA. The results of the biofilm inhibition assay indicated that TCA can significantly inhibit the biofilm formation and the expression of the biofilm regulatory gene *hld* [14]. The biofilm of MRSA has serious clinical implications, making it difficult to eradicate and more tolerant to antibiotic therapy [15]. When the formation of biofilm was inhibited, the resistance of MRSA to antibiotics can be reduced, which may be one of the reasons that TCA combined with antibiotics has a synergistic effect. In addition, it has been reported that TCA has a hydrophobic structure that can change the permeability of cell membranes and destroy the integrity of cell membranes and cell walls [16]. We speculated that the efficiency of antibiotics entering into cells increased as TCA destroyed the cell membrane, implying that TCA improved the sensitivity of antibiotics to bacteria.

TCA is an essential oil derived from cinnamon bark, with certain limitations, such as low water solubility and low stability. Recent studies have demonstrated that the encapsulation of essential oil by nanomaterials can increase chemical stability and solubility [17], equipping TCA with promising clinical significance as an antibiotic reversal agent. In conclusion, our results indicated that TCA exhibited a potent in vitro ability to restore antibiotic sensitivity against MRSA at sub-MIC, revealing the promising potential of TCA to overcome the antibiotic resistance crisis and be developed to a useful botanical synergist of antibiotics for the control of MRSA. However, the study has certain limitations. First, the *S. aureus* used in the study only one reference strain and two clinical isolates, which were confined to a selected population from a single hospital in Korea with a specific endemic [4]. Moreover, it may limit the external validity of the test results. Second, TCA is a food-grade compound. Nevertheless, there may be potential side effects when it is used in combination with antibiotics. Their clinical usage needs to be determined using in vivo efficacy studies.

In addition to discovering novel antibacterial treatments, implementing interventions to control the spread of MRSA in the community and improve compliance with basic infection control measures also have crucial significance for further community-associated MRSA transmission and infections [18].

## 4. Materials and Methods

### 4.1. Reagents

TCA, crystal violet, ampicillin, oxacillin, gentamicin, vancomycin, amoxicillin, ceftazidine, amikacin, and cefoxitin were purchased from Sigma-Aldrich Co. (St. Louis, MO, USA). Mueller-Hinton agar, Mueller-Hinton broth, and skim milk were obtained from Difco Laboratories (Baltimore, MD, USA). Primary antibodies to PBP2a were obtained from DiNonA Inc. (Seoul, Korea). Anti-mouse IgG secondary antibody was obtained from Thermo Fisher Scientific Inc. (Carlsbad, CA, USA). The chemiluminescent ECL assay kit was obtained from ATTO Corp. (Tokyo, Japan). E.Z.N.A. Bacterial RNA Kit was purchased from Omega Bio-Tek (Norcross, GA, USA). The sequences of primers used in this study were obtained from Bioneer (Daejeon, Korea) (Table 2). SMART™ bacterial protein extraction solution was obtained from Intron BioTechnology, Inc. (Seongnam, Korea). QuantiTect Reverse Transcription Kit was obtained from (Dusseldorf, Germany). Power SYBR Green PCR Master Mix was obtained from Life Technologies LTD (Warrington, UK).

### 4.2. Susceptibility Testing of TCA with Conventional Antibiotics

The MIC values of TCA were determined either alone or in combination with conventional antibiotics (ampicillin, oxacillin, gentamicin, vancomycin, amoxicillin, ceftazidine, amikacin, and cefoxitin) according to the guidelines of the Clinical and Laboratory Standards Institute (CLSI) by microdilution and checkerboard assays [19]. *S. aureus* strains were cultured on MHA at 37 °C for 24 h. Serial dilutions of TCA with antibiotics were mixed in Mueller Hinton broth (MHB). The MRSA inocula were adjusted to 0.5 McFarland standard in MHB. The bacterial concentration of the final inoculum was 1.5 × 10^5^ CFU/well. Each MIC value was determined after a 24 h incubation period at 37 °C and defined as the lowest concentration that completely inhibited the growth of the bacteria. In vitro interaction between the drugs was quantified by determining the fractional inhibitory concentration index (FICI), as follows: ∑FIC: FICA + FICB = MICA + B/MICA alone + MICB + A/MICB alone. The combination was considered as synergy for FICI ≤ 0.5, partial synergy for 0.5 < FICI ≤ 0.75, additive effect for 0.75 < FICI ≤ 1, indifferent for 1 < FICI ≤ 4 and antagonism for FICI > 4. At the same time, the fold reduction of the MIC of antibiotics against MRSA alone to combined with TCA also was calculated, which is abbreviated as Fold in Table 1. All tests were performed in triplicate.

### 4.3. Bacterial Strains and Conditions

*S. aureus* ATCC 33591 (American Type Culture Collection, Manassas, VA, USA) was used as a reference strain. Two clinical isolates of MRSA DPS-1 and DPS-3 were isolated from patients at the Wonkwang University Hospital (Jeonbuk, Korea). Bacteria were maintained on Mueller–Hinton agar (MHA) plates. Liquid cultures for *S. aureus* strains were cultured in Mueller–Hinton broth (MHB) at 37 °C.

### 4.4. Time–Kill Assays 

The synergistic antimicrobial effect was further determined by a time–kill assay, which used 1/2 MIC of TCA and 1/2 MIC of three conventional antibiotics (amikacin, gentamicin, and oxacillin) against the growth *S. aureus* (reference strain ATCC33591 and clinical isolates DPS-1, DPS-3) according to the previous method [10]. Briefly, bacteria cultures incubated in MHA at 37 °C for 24 h were diluted with sterilized MHB to 1.5 × 10^5^ CFU/mL, and the diluted cultures were incubated at 37 °C for 24 h. Then, aliquots (100 µL) of the culture were removed after an incubation period at (0, 4, 8, 16, and 24 h), and serial 10-fold dilutions were prepared in saline as needed. The numbers of viable cells were counted on a drug-free MHA plate after 24 h incubation. Three independent experiments were carried out and the data presented as the mean ± SD. Finally, the graphics as log10 CFU/mL versus time were plotted to describe the results of time–kill assays, and the combinations decrease of CFU/mL by ≥2 log10 was considered as a synergistic effect [20].

### 4.5. Western Blot Analysis

Western blot analysis was carried out to evaluate the effect of TCA on the expression of PBP2a according to the methods described previously [21]. *S. aureus* strains (ATCC 33591) were grown in MHB for 24 h and treated with sub-concentrations of TCA for 30 min. Cell protein extracts were harvested by centrifugation at 3000× *g* for 10 min and the protein concentration was determined by a Bio-Rad protein assay reagent (Bio-Rad Laboratories, Hercules, CA, USA). The supernatants were subjected to SDS-PAGE and electroblotted onto Amersham HybondTM-P membranes (GE Healthcare, Piscataway, NJ, USA). The membranes were blocked by 5% skim milk and probed with monoclonal mouse anti-PBP2a primary antibody (diluted 1:1000, DiNonA, Seoul, Korea) overnight at 4 °C and re-probed with anti-mouse IgG secondary antibody (diluted 1:2000, Enzo Life Sciences, Ann Arbor, MI, USA) at room temperature for 2 h. Then, the membranes were supplemented with ECL Prime Western Blotting Detection reagent (GE Healthcare Life Sciences, Incheon, Korea), and an ImageQuant LAS-4000 mini chemical luminescent imager (GE Healthcare Life Sciences) was used to visualize the bands [22].

### 4.6. qRT-PCR

*S. aureus* ATCC 33591 in MHB was treated with sub-inhibitory concentrations (1/8 MIC, 1/4 MIC, and 1/2 MIC) of TCA for 0.5 h with untreated as a control. Easy-RED BYF RNA was prepared with the Easy-RedByFrNA extraction kit according to the manufacturer’s instructions (iNtRON Biotechnology, Seongnam, Korea). RNA was reverse transcribed into cDNA and was performed by a cDNA synthesis kit (iNtRON Biotechnology) for first-strand cDNA synthesis, in accordance with the manufacturer’s instructions, in order to synthesize the RNA template for qRT-PCR. Primers used are presented in Table 1. A total 20 μL volume was used in the qRT-PCR: 2 μL sample cDNA and 2 μL of primer mix (10 μM), 10 μL of 2 × SYBR premix (Life Technologies, Carlsbad, CA, USA), and 6 μL of nuclease-free water. The PCR was performed by the StepOnePlus real-time PCR system (Applied Biosystems, Courtaboeuf, France).

### 4.7. Biofilm Inhibition Assay

The effect of the TCA on the biofilm formation of *S. aureus* (ATCC 33591 and DPS-1) had been identified by biofilm inhibition assay as previously described [9]. One hundred microliters of overnight cultures (0.5 MacFarland bacterial culture) were added and treated with various concentrations of TCA (1/16 MIC, 1/8 MIC, and 1/2 MIC) to each well of 96-well microtiter plates. After incubation for 24 h at 37 °C, the planktonic cells were removed, washed with PBS for 3 times in each well, stained with 1% (*w*/*v*) crystal violet for 10 min, and then washed again with PBS. The stained biofilms were solubilized in 100 μL of absolute ethanol and the optical density (OD) values were measured at 600 nm. All biofilm assays were performed in triplicates, and the negative control was the bacteria in their respective media without any drug. Finally, the percentage of inhibition of biofilm was calculated by the following formula. Percentage of inhibition = 100 − [(OD 600 nm of the treated wells)/(mean OD 600 nm of the negative control wells) × 100)].

### 4.8. Statistical Analysis

Analyses were performed in triplicate and data were presented as the mean ± standard deviation. The results were statistically analyzed by an independent Scheffe’st-test (SPSS software version 22.0; IBM SPSS, Armonk, NY, USA). A *p*-value of less than 0.05 was considered to be statistically significant.

## Figures and Tables

**Figure 1 ijms-22-02752-f001:**
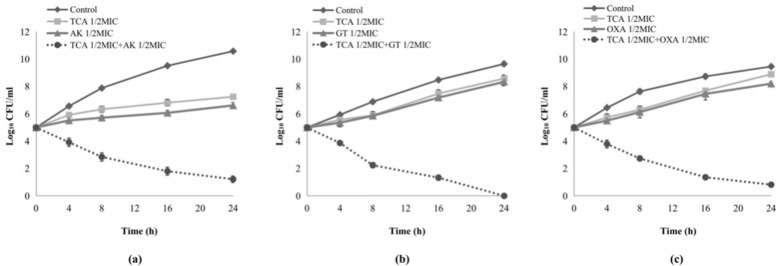
Time–kill curves of the synergistic effect of TCA in combination with three conventional antibiotics at sub-inhibitory concentrations against methicillin-resistant *Staphylococcus aureus* (MRSA). (**a**) Time–kill curves of combination between TCA and amikacin against reference strain ATCC 33591. (**b**) Time–kill curves of combination between TCA and gentamicin against clinical isolates DPS-1. (**c**) Time–kill curves of combination between TCA and oxacillin against clinical isolates DPS-3. TCA: trans-cinnamaldehyde; AK; CFU, colony-forming units; MIC, minimum inhibitory concentration.

**Figure 2 ijms-22-02752-f002:**
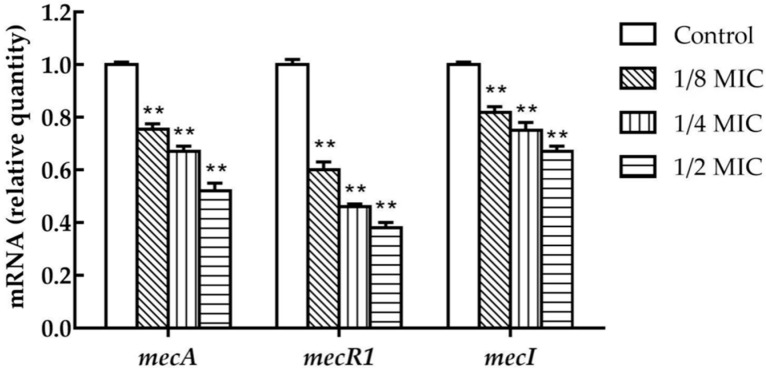
The expression of *mecA*, *mecR1*, and *mecI* in MRSA (ATCC 33591) cultures in the presence of sub-concentrations of TCA. The relative gene expression of *mecA*, *mecR1*, and *mecI* was reduced in a dose-dependent manner. The data were presented as the mean ± standard deviation of the three independent experiments. ** represents *p* < 0.01.

**Figure 3 ijms-22-02752-f003:**
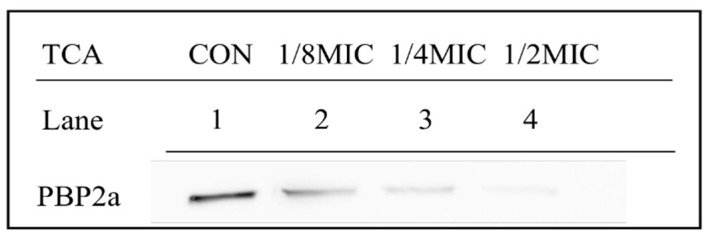
Effect of TCA at sub-concentrations on the expression of penicillin-binding protein 2a (PBP2a) in MRSA (ATCC 33591). Lane 1, the control was treated without TCA; Lane 2-4, PBP2a production after treatment with TCA at 1/8 MIC, 1/4 MIC, and 1/2 MIC, respectively.

**Figure 4 ijms-22-02752-f004:**
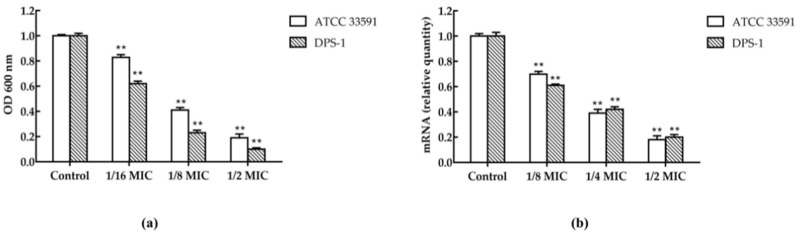
(**a**) Inhibition of TCA at sub-inhibitory concentrations on MRSA (ATCC 33591 and DPS-1) biofilm. (**b**) The expression of *hld* in MRSA (ATCC 33591 and DPS-1) cultures in the presence of TCA at various concentrations. The data were presented as the mean ± standard deviation of the three independent experiments. ** represents *p* < 0.01.

**Table 1 ijms-22-02752-t001:** The MIC value of conventional antibiotic and synergistic effect of TCA combined with an antibiotic.

Antibiotics	ATCC 33591			DPS-1			DPS-3		
	MIC (μg/mL)	Fold	FICI	MIC (μg/mL)	Fold	FICI	MIC (μg/mL)	Fold	FICI
Ampicillin	62.5	2	1	31.3	4	0.75	62.5	4	0.75
Oxacillin	62.5	4	0.75	500	2	1	500	8	0.25
Gentamicin	3.9	4	0.75	125	4	0.37	250	4	0.5
Vancomycin	250	8	0.25	250	2	1	500	2	1
Amoxicillin	62.5	8	0.63	125	4	0.5	125	8	0.25
Ceftazidine	125	2	1	125	2	1	250	4	0.75
Amikacin	31.2	4	0.38	31.2	8	0.25	62.5	16	0.19
Cefoxitin	31.2	4	0.75	62.5	8	0.62	250	4	0.5

TCA, trans-cinnamaldehyde; MIC, minimal inhibitory concentration; Fold, Fold reduction of the MIC of antibiotics. FICI, The fractional inhibitory concentrations index.

**Table 2 ijms-22-02752-t002:** Primers used in qRT-PCR.

Primer	Sequence (5′-3′)
*16S*	F: ACTCCTACGGGAGGCAGCAG
	R: ATTACCGCGGCTGCTGG
*mecA*	F: CAATGCCAAAATCTCAGGTAAAGTG
	R: AACCATCGTTACGGATTGCTTC
*mecR1*	F: GTGCTCGTCTCCACGTTAATTCCA
	R: GACTAACCGAAGAAGTCGTGTCAG
*mecI*	F: CGTTATAAGTGTACGAATGGTTTTTG
	R: TCATCTGCAGAATGGGAAGTT
*hld*	F: ATTTGTTCACTGTGTCGATAATCC
	R: GGAGTGATTTCAATGGCACAAG

## Data Availability

Personal information is included, so it was conducted for research only.

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
