# Peer review of "Trans-Cinnamaldehyde Exhibits Synergy with Conventional Antibiotic against Methicillin-Resistant Staphylococcus aureus"

_ijms, 2021, doi:10.3390/ijms22052752_

Round 1

Reviewer 1 Report

The manuscript reports on the synergistic antibacterial action demonstrated by trans-cinnamic acid with a series of commercially available antibiotics. The main outcome of the manuscript relies upon a number of biological assays, from which the Authors briefly conclude about the possible mechanism od synergistic action.

After reading the manuscript  I have an ambivalent feelings.

Undoubtedly, the research was carried out in a proper manner, the results are provided in a clear and interesting way, and the whole work is properly discussed. Certainly the results produced by the authors are worth publishing.

The main doubt, however arises from the scope of the journal to which this work is submitted. Also, the content of the manuscript is limited only to a number of biological assays, while usually the IJMS articles consist of extensive spectroscopic, and mathematical data. In my opinion this manuscript does not quite fit the scope of IJMS, and I would strongly recommend its transfer to a more appropriate Journal such as MDPI Antibiotics, Biomolecules, or Pharmaceuticals.  

Reviewer 2 Report

The manuscript entitled “Trans-cinnamaldehyde Combination Conventional Antibiotic Therapy for Methicillin-resistant Staphylococcus aureus” by Wang et al. describes the in vitro antimicrobial effect of the combination of TCA with several antibiotics against MRSA. In my opinion this research is worth publishing, since the authors have proved the great effectiveness of some combinations, this opening interesting possibilities to reduce the amount of antibiotics used. Nevertheless, I would point out some improvements that could be made to the text in order to make it more easily understood.

Although I am not an English speaker, I would suggest to have the manuscript revised at the grammatical level, since some sentences could be improved. Some sentences would need rephrasing.

Line 2. The title of the manuscript itself is grammatically incorrect.

Line 55. Table 2 must be Table 1

Line 55. The authors write the acronym FICI, without having mentioned its meaning (fractional inhibitory concentrations index).

Table 1.  It should be specified if MIC refers to the antibiotic alone or in combination with TCA.

Line 164. “encapsulation of nanomaterials by essential oil” must be the opposite: “encapsulation of essential oil by nanomaterials”

Line 208. “three conventional antibiotics”. But what about the other 5? Weren´t they tested in the same way? Although the authors only show graphics for three antibiotics, table 1 provides data for combinations with 8 antibiotics.

Reviewer 3 Report

Authors have presented a very nice work that presents important topic in the times of increasing microbial resistance. They have written a clear balanced text a have presented the results very clearly.

I have just few comments to the manuscript:

Line 55: It should be Table.1  instead of Tab2.(Tab 2.-  Describes the used primers).

To the methodology I have few comments:

  1. Authors are describing that the sensitivity was assayed according the CLSI standard method. Are the MIC values really the concentration that are inhibiting the bacterial growth for 100% or 50% it is not clear if it is only the 50% inhibition- it should be mentioned in the text or indexed as MIC50. If it is really the 100% inhibition it is ok.

My main comment is to the Western Bolt results. How did the authors normalized the proteins for the western blot? There is no blot for the housekeeping  protein. Could the authors explain it or add the protein housekeeper blot  to the blot of the PBP2a.

Authors could discuss little more the mechanism of the synergism with gentamicin and amikacin, here the decreased PBP2a  does not explain the synergy.

I have no other comments to the text.
